# A Benchmark of In-House Homologous Recombination Repair Deficiency Testing Solutions for High-Grade Serous Ovarian Cancer Diagnosis

**DOI:** 10.3390/diagnostics13213293

**Published:** 2023-10-24

**Authors:** Rodrigo Guarischi-Sousa, José Eduardo Kroll, Adriano Bonaldi, Paulo Marques Pierry, Darine Villela, Camila Alves Souza, Juliana Santos Silva, Matheus Carvalho Bürger, Felipe Azevedo Oliveira, Marcelo Gomes de Paula, Fabiana Marcelino Meliso, Luiz Gustavo de Almeida, Priscilla Morais Monfredini, Ana Gabriela de Oliveira, Fernanda Milanezi, Cristovam Scapulatempo-Neto, Guilherme Lopes Yamamoto

**Affiliations:** Diagnósticos da América S.A. (DASA), São Paulo 06455-010, Brazil; jose.kroll.ext@dasa.com.br (J.E.K.); adriano.bonaldi@dasa.com.br (A.B.); paulo.pierry@dasa.com.br (P.M.P.); darine.villela.ext@dasa.com.br (D.V.); calves.souza@dasa.com.br (C.A.S.); julianass@dasa.com.br (J.S.S.); matheus.burger@dasa.com.br (M.C.B.); felipe.oliveira@dasa.com.br (F.A.O.); marcelo.paula@dasa.com.br (M.G.d.P.); fabiana.marcelino@dasa.com.br (F.M.M.); luiz.dufner@dasa.com.br (L.G.d.A.); priscilla.morais@dasa.com.br (P.M.M.); anagabriela.oliveira@dasa.com.br (A.G.d.O.); fernanda.milanezi.ext@dasa.com.br (F.M.); cristovam.neto@altadiagnosticos.com.br (C.S.-N.); guilherme.yamamoto.ext@dasa.com.br (G.L.Y.)

**Keywords:** homologous recombination deficiency (HRD), biomarkers, ovarian cancer, DNA repair, *BRCA1*/*2*, poly (ADP–ribose) (PARP) inhibitors (PARPi), next generation sequencing (NGS)

## Abstract

Homologous recombination deficiency (HRD) has become an important prognostic and predictive biomarker for patients with high-grade serous ovarian cancer who may benefit from poly-ADP ribose polymerase inhibitors (PARPi) and platinum-based therapies. HRD testing provides relevant information to personalize patients’ treatment options and has been progressively incorporated into diagnostic laboratories. Here, we assessed the performance of an in-house HRD testing system deployable in a diagnostic clinical setting, comparing results from two commercially available next-generation sequencing (NGS)-based tumor tests (SOPHiA DDM^TM^ HRD Solution and AmoyDx^®^ (HRD Focus Panel)) with the reference assay from Myriad MyChoice^®^ (CDx). A total of 85 ovarian cancer samples were subject to HRD testing. An overall strong correlation was observed across the three assays evaluated, regardless of the different underlying methods employed to assess genomic instability, with the highest pairwise correlation between Myriad and SOPHiA (R = 0.87, *p*-value = 3.39 × 10^−19^). The comparison of the assigned HRD status to the reference Myriad’s test revealed a positive predictive value (PPV) and negative predictive value (NPV) of 90.9% and 96.3% for SOPHiA’s test, while AmoyDx’s test achieved 75% PPV and 100% NPV. This is the largest HRD testing evaluation using different methodologies and provides a clear picture of the robustness of NGS-based tests currently offered in the market. Our data shows that the implementation of in-house HRD testing in diagnostic laboratories is technically feasible and can be reliably performed with commercial assays. Also, the turnaround time is compatible with clinical needs, making it an ideal alternative to offer to a broader number of patients while maintaining high-quality standards at more accessible price tiers.

## 1. Introduction

High-grade serous ovarian cancer is the most common type of ovarian cancer and the leading cause of death in women diagnosed with gynecological malignancies; it appears to originate from the epithelial tissue, arising primarily from the ovaries, fallopian tubes, or peritoneum [1,2,3]. Following standard treatment approaches of cytoreductive surgery and platinum-based therapy, the 5-year survival rate is approximately 30% [4,5]. Next-generation sequencing (NGS) has enabled systematic investigations of genomic and molecular alterations that drive ovarian cancer, aiming to identify patients who may respond to targeted therapies [6,7]. Notably, genomic instability is a hallmark in the formation of tumors, and inefficient DNA repair is a critical driving force behind cancer establishment, progression, and evolution [8].

Homologous recombination repair (HRR) is a DNA repair pathway that acts on DNA double-strand breaks and interstrand cross-links [9,10,11,12]. Many tumor types, including ovarian cancer, exhibit a deficiency in the HRR pathway; such a phenotype has been termed homologous recombination deficiency (HRD) [13], and it has been extensively shown that the presence of HRD can make tumors present a distinct clinical constitution involving a superior response to platinum-based therapies and poly (adenosine diphosphate [ADP]-ribose) polymerase (PARP) inhibitors (PARPi) [14,15,16]. The introduction of PARPis has transformed the management of high-grade serous ovarian cancer in both relapsed and first-line treatment settings [17,18]. Developing methods to reliably determine HRD status is of critical importance to optimizing clinical benefit from these drugs. Thus, diagnostic companies have developed assays to determine HRD status and aid in treatment decisions; however, these assays may differ in what they evaluate and may lead to inconsistent results that can be complicated for prescribing physicians [19]. In general, the assays have been developed to measure the causes and/or consequences of the HRD phenotype.

The best-characterized causes of HRD in ovarian cancer are germline or somatic mutations in *BRCA1* and *BRCA2* [20]. Nonetheless, there is now clear evidence that HRD can arise through germline and somatic mutations or even methylation of a wider set of homologous recombination repair (HRR)-related genes [21,22]. Further characterization of the HRR pathway has revealed multiple protein co-factors that are necessary for functional HRR, and mutations in the genes that encode these proteins could also contribute to ovarian cancer [21,22]. Regarding testing the consequences of the HRD phenotype, the assays are performed to estimate the genomic instability (genomic “scars”) or mutational signatures that measure the patterns of somatic mutations accumulating in HRD cancers irrespective of the underlying defect. Numerous studies in ovarian cancer have identified mutational signatures of instability associated with an HRD phenotype, including loss of heterozygosity (LOH), telomeric imbalances (TAI), and large-scale transitions (LST) [23,24,25].

Although there are some diagnostic tests commercially available to promote tumor selection for personalized therapy based on HRD status [19], the Myriad MyChoice CDx is the most notorious FDA-approved tumor assay to assess chromosomal instability because it incorporates both the causes and consequences of HRD status [26,27], while other assays only detect potential causes of HRD status without assessing the consequences. However, this test is still vastly limited in that it is locally performed, highly priced, and frequently not reimbursed by medical insurance. As an alternative, other commercial assays deployable in diagnostic laboratories that screen for both HRD status and genomic instability were recently launched (SOPHiA DDM^TM^ HRD Solution and AmoyDx HRD Focus Panel), even though the main challenge is a lack of standardized methods to define, measure, and report HRD status. Hence, conducting studies that assess the correlation of HRD status across different assays is extremely important since it can identify sources of discordance, providing an opportunity to determine the underlying causes of inconstancy and providing information for better use of these assays for cancer treatment decisions. In this study, we aimed to examine the technical viability of in-house HRD testing, particularly for ovarian cancer diagnosis, by comparing the results of two commercially available NGS-based tumor tests to the reference assay Myriad MyChoice CDx.

## 2. Materials and Methods

### 2.1. Tumor Samples

The tumor DNA samples selected for this study were obtained from 85 patients with ovarian cancer referred for genetic testing upon medical request in our laboratory. High-grade serous ovarian cancer samples originating from distinct primary tissues such as the ovaries, fallopian tubes, or peritoneum were retrieved from the pathology department. Archival histological sections of 10 µm from formalin-fixed paraffin-embedded (FFPE) tumor samples were obtained, and a hematoxylin and eosin-stained slide was reviewed by pathologists to select and mark a representative tumor area for macrodissection and DNA extraction. The clinical and pathological features of all samples used in this technical validation study are presented in Table 1 and Appendix A.

### 2.2. HRD Testing 

In-house HRD testing assessment was conducted comparing two commercial kits, SOPHiA DDM^TM^ Homologous Recombination Deficiency (HRD) Solution and AmoyDx HRD Focus Panel, to the reference assay Myriad MyChoice CDx. Despite the different methodologies, both commercial kits allow simultaneous detection of *BRCA* status and predict a genomic instability score. While MyChoice CDx testing was performed externally, with FFPE slides being sent to a reference laboratory to perform all analytical steps, the other two methods had DNA extraction, library preparation, sequencing, data analysis, and clinical interpretation performed internally. Regions with the highest tumor cell density, previously marked by a pathologist, were scraped from five to eight slides for each sample. DNA was extracted using the Maxwell RSC DNA FFPE kit (Promega, Madison, WI, USA) and quantified using two methods: Qubit™ dsDNA High Sensitivity (Thermo Fisher Scientific, Waltham, MA, USA) and/or Real-Time Quantitative PCR, using the TaqMan™ Fast Advanced Master Mix (Thermo Fisher Scientific) and the TaqMan^®^ Copy Number Assays (Thermo Fisher Scientific).

#### 2.2.1. SOPHiA DDM^TM^ HRD Solution 

DNA library preparation was performed using the SOPHiA DNA Library Prep Kit II (SOPHiA GENETICS, Lausanne, Switzerland), following the manufacturer’s recommendations. DNA inputs ranged from 3.9 to 150 ng according to the functional concentration measured by real-time PCR. The fragmentation time was adjusted based on the degradation level of the tumor sample, analyzed using automated electrophoresis separation with Genomic DNA ScreenTape (Agilent Technologies, Santa Clara, CA, USA). Sequencing libraries were quantified using the Qubit™ dsDNA High Sensitivity Assay kit (Thermo Fisher Scientific), and automated electrophoretic separation was performed to analyze the fragment size distribution using the D1000 DNA ScreenTape (Agilent Technologies, Santa Clara, CA, USA). HRD testing from SOPHiA GENETICS involves a homologous recombination repair (HRR) multi-gene panel and low-pass whole genome sequencing (LP-WGS) that were carried out independently in two steps. First, the HRR panel evaluates single nucleotide variants (SNVs), small insertions and deletions (InDels), and copy number variations (CNVs) in the coding regions and splicing sites of the following genes: *ATM*, *BARD1*, *BRCA1*, *BRCA2*, *BRIP1*, *CDK12*, *CHEK1*, *CHEK2*, *FANCL*, *PALB2*, *PPP2R2A*, *RAD51B*, *RAD51C*, *RAD51D*, *RAD54L*, and *TP53*. The HRR panel was sequenced in the MiSeq System from Illumina Inc. (San Diego, CA, USA) using the MiSeq Reagent Kit V3 (600 cycles in a Paired-End 2 × 151 cycles run). The limit of detection (LOD) for SNVs and InDels was 5%, and the minimal depth of coverage was 500× (96.05% sensitivity and 99.99% specificity). Importantly, for the evaluation of HRD status in ovarian cancers, it has been demonstrated that only pathogenic or likely pathogenic variants in *BRCA1/2* are relevant to current literature data; therefore, despite the other genes in the HRR panel, only variants on these two genes were considered in this study. Second, the LP-WGS experiments were performed either on the NextSeq 550 or NovaSeq 6000 sequencing platforms from Illumina Inc., using the NextSeq Mid Output Reagent kit (300 cycles in a Paired-End 2 × 151 cycles run) or the NovaSeq S1 Reagent kit v1.5 (200 cycles in a Paired-End 2 × 101 cycles run), respectively. The targeted vertical coverage during sequencing is 1–3×. The primary sequencing output was demultiplexed by bcl2fastq v2.20.0.422, and the reads were processed to trim adapters and low-quality base calls. Data analysis was carried out using the SOPHiA DDM^TM^ platform, in which the reads were mapped to the human genome (hg19) and sequencing depth was computed and normalized to calculate a genome-wide coverage profile. A convolutional neural network algorithm previously trained based on data from the Cancer Genome Atlas Program (TCGA) [28] takes as input a bitmap-like matrix that reflects the whole genome sequencing coverage profile and outputs a scalar value of the Genomic Integrity Index (GII). Tumor samples with scores above the 1.8 threshold indicate a GII positive status, reflecting an inability to repair double-strand breaks, whereas scores below that threshold indicate a GII negative status for which PARPi therapy would not be recommended. The GII score had a non-strict range of about −30 up to +30.

#### 2.2.2. AmoyDx HRD Focus Panel

DNA library preparation was performed using the AmoyDx HRD Focus Panel kit (AmoyDx, Xiamen, China), following the manufacturer’s recommendations. The assay was intended to determine HRD status via the detection of pathogenic variants in the *BRCA1/2* genes and the determination of a genomic instability score in the tumor samples in a single workflow. The assay kit is based on the Halo-shape ANnealing and Defer-Ligation Enrichment (HANDLE) system technology, in which the probes contain an extension and ligation arms complementary to the target gene regions. DNA inputs ranging from 50 to 100 ng were pre-denatured, and the probes hybridized to the target regions of *BRCA1/2*. Then, an extension and ligation step of the probes were performed, treated with exonuclease for the digestion of non-hybridized probes and free single- and double-stranded DNAs in the solution. The linked probes were amplified by PCR, and the final product was purified. Libraries were quantified with Qubit™ before and after the purification step to verify material losses. A total of 100 ng of each library were polled together, and the final concentration was measured on Qubit™. DNA libraries were sequenced on the NovaSeq 6000 platform from Illumina Inc. using the NovaSeq SP Reagent Kit v1.5 (300 cycles in a Paired-End 2 × 150 cycles run). In brief, the detection of pathogenic variants was conducted within the coding sequence and at the boundaries of the exon-intron of the *BRCA1/2* genes. The estimated Genomic Scar Score (GSS) was calculated by evaluating 24,000 SNPs, where a GSS equal to or higher than 50 is suggestive of a positive HRD status. The GSS model from AmoyDx is built on machine learning, and it measures genomic instability by weighing distinct types of chromosomal copy numbers, which were grouped according to the combination of length, type, and site of the copy number. Data analysis was carried out using the ANDAS System (AmoyDx, China) with pipeline version 1.1.1.

#### 2.2.3. Variant Classification

Detected variants in *BRCA1* and *BRCA2* were classified for their clinical impact according to the American College of Medical Genetics (ACMG) [29], Somatic Oncogenicity SOP—ClinGen (https://clinicalgenome.org) [30], and CanVIG-UK (https://www.cangene-canvaruk.org) [31] guidelines. Only variants classified as pathogenic (P), likely pathogenic (LP), and variants of unknown significance (VUS) were reported. 

### 2.3. Data Analysis and Visualization

Comparative analysis was conducted on Python 3 with the packages Pandas [32] and SciPy [33]. Results were considered statistically significant if the *p*-value < 0.05. Data visualization was performed using the packages Seaborn [34] and Matplotlib [35].

## 3. Results

### 3.1. Clinicopathological Features of Tumor Samples

Our evaluation involved the comparison of *BRCA1/2* status and predicted genomic instability (PGI) score with the final assigned sample status obtained from the assays. For this purpose, we utilized a dataset of 85 ovarian carcinoma samples (Table 1). The median age of the patients during the sample collection was 60 years old. It is important to note that the FFPE samples did not exhibit uniform levels of DNA input, neither in quantity nor quality. The estimated tumor content ranged from 20% to over 90%, with DNA input varying from 3.9 to 150 ng. Appendix A presents all relevant information about the samples included in this study, including the pre-analytical variables assessed in our analysis.

### 3.2. Comparison of Predicted Genomic Instability (PGI) Score across Different NGS-Based Methods 

Although the three assays evaluated employ different underlying methods to assess genomic instability in tumor samples, all of them provide scalar values that reflect the PGI score (i.e., higher values indicate greater genomic instability, while lower values indicate lesser instability). We investigated the associations and discrepancies among the methods, and we found an overall strong correlation, with the highest pairwise correlation observed between Myriad and SOPHiA (R = 0.87, *p*-value = 3.39 × 10^−19^; Figure 1A). It is important to note that the interval ranges and the proportion of failed samples vary between the methods. Also, not all samples could be evaluated using the three methodologies due to insufficient tumor material or DNA quantity. Still, a total of 32 samples were executed across the three assays, of which, for SOPHiA’s test, 96.9% (31 samples) passed the quality control (QC) criteria. One sample was flagged as having a “silent profile”, suggesting either a genome-wide profile without copy number variations (CNVs) or low tumor content. For Amoy’s test, 84.4% (27 samples) passed the QC filter, and 5 samples failed the QC criteria (Appendix A). This finding is consistent with the 84.2% success rate reported by another group that also evaluated the same commercial assay [36].

We subsequently evaluated the level of agreement in assigned sample statuses based on the pre-defined thresholds provided by the vendors, specifically 1.8 for SOPHiA and 50 for AmoyDx. In the case of SOPHiA’s test, we observed categorical discrepancies in three samples when compared to the results obtained from Myriad’s test. Specifically, 2 samples initially assigned as negative and 1 sample initially assigned as positive were predicted to have the opposite status. Notably, these samples exhibited scores near the threshold values of both tests, highlighting the need for careful consideration of their categorical classification. Contrarily, we identified 7 categorical discrepancies between Myriad and AmoyDx, in which 6 samples were initially assigned as being negative by Myriad but were predicted to be positive by AmoyDx (Figure 1B).

### 3.3. Correlation between BRCA1/2 Status and Predicted Genomic Instability (PGI) Score

Considering that a high degree of genomic instability is commonly observed in *BRCA*-mutated tumors [37,38], we investigated the association and consistency between these two markers in a set of 26 samples in which we successfully executed all three methodologies and passed the quality QC criteria. All samples with a detected pathogenic variant in *BRCA1/2* were positive for genomic instability; the only exception was one sample with a pathogenic variant in *BRCA1* that was categorized as negative for genomic instability in the AmoyDx test. 66.7% (8 of 12) of BRCA-positive samples were also HRD-positive for Myriad, 57.1% (8 of 14) for SOPHiA, and 43.8% (7 of 16) for AmoyDx. As all pathogenic and likely pathogenic variants from Myriad’s results were confirmed in the other two tests, this small association deflation between *BRCA1/2* status and PGI score is due to more samples having a PGI-positive score in the later assays. Noteworthy, the SOPHiA test was able to detect one somatic variant in *BRCA1* with a variable allele fraction (VAF) of 9.8%, not previously reported by the other two tests (Figure 2).

### 3.4. Determination of Testing Performance Parameters

Controlling experimental variability is crucial in any clinical test. NGS-based assays, being a multi-step process, are particularly susceptible to variations that may arise from extraction, library preparation, and sequencing, as these steps involve non-deterministic processes. Even prior to these steps, the extraction of DNA from somatic specimens relies on a biopsy process that samples a heterogeneous population of cells. To evaluate the reproducibility of our experiments, we sequenced seven unique samples as replicates within or between sequencing runs. Among them, 4 samples were sequenced across different sequencing batches, while one sample was sequenced within the same batch for SOPHiA and 3 samples within the same batch for AmoyDx (Figure 3). Remarkably, we observed a high correlation of results for all tested samples: for SOPHiA, r = 0.998 (*p*-value = 1.17 × 10^−4^), and for AmoyDx, r = 0.995 (*p*-value = 0.064; Figure 3). These results highlight the consistency and reliability of the obtained data, indicating robust reproducibility within our experimental setup.

In respect to the comparison of HRD status (i.e., final assigned sample result) of the two assays to the reference Myriad’s test, the positive predictive value (PPV) and negative predictive value (NPV) were 90.9% and 96.3% for SOPHiA’s test, respectively, while AmoyDx’s test achieved 75% PPV and 100% NPV. In our cohort, we identified 44.9% and 64.9% as HRD-positive tumors using SOPHiA and AmoyDx, respectively (Table 2).

### 3.5. Association of Pre-Analytical Variables with Low Confidence NGS Results

Pre-analytical variable verification is a crucial step to ensure a reliable experiment result, and it can also assist in the prediction of samples that are likely to pass or fail during the QC step. Both in-house HRD testing protocols evaluated suggest at least 30% estimated tumor content but have slightly different ranges for DNA input requirements, between 10–200 ng for SOPHiA and 50–100 ng for AmoyDx. Among the four pre-analytical variables evaluated, we observed that only functional DNA qPCR and DNA input were statistically significant between pass and low-confidence experiments (*p*-value = 3.52 × 10^−4^; *p*-value = 3.40 × 10^−4^) (Figure 4A). However, we observed a negative correlation between DNA library yield and residual noise (R = −0.62, *p*-value = 1.72 × 10^−9^), which is computed by measuring the standard deviation of the normalized genome sequencing coverage profile with respect to the smoothed normalized genome sequencing coverage profile (Figure 4B). Indeed, it is expected that samples with low values of library yield have more fragmented DNA, resulting in potential amplification biases and less uniform coverage in the genome, making the analysis of those experiments more challenging. Figure 4C shows two examples of samples that have low and high residual noise profiles.

## 4. Discussion

HRD testing provides relevant information to personalize patients’ treatment options, and it has been progressively incorporated into diagnostic laboratories. Although the Myriad MyChoice^®^ CDx assay was clinically validated and became a reference in the market, unfortunately, the test is not accessible to all patients. Besides being expensive and not reimbursed, a centralized assay poses significant challenges, such as the considerable rejection rate due to the strict quality requirements of externalized technical procedures and the increased turnaround time. Thus, considering the clinical need for a lower-cost, highly accurate, and reduced laboratory turnaround time, other assays were recently launched to determine HRD status. However, the methodologies are different, and the uncertainty in how to measure and report HRD status can potentially lead to low adoption of this test in clinical routine. To simplify technical procedures workflows and data analysis interpretation, many attempts have been made by several medical centers to employ in-house HRD testing. Here, we report our experience implementing HRD testing in a clinical diagnostic setting. We evaluated the feasibility of in-house HRD testing by comparing the results of two commercially available NGS-based tumor tests to the reference assay Myriad MyChoice^®^ CDx. The use of a clinically representative dataset of tumor samples allowed a systematic assessment of the detection power and concordance rate of HRD status across the assays, identifying sources of inconstancy and providing information for better use of these assays for clinical decision-making.

In the clinical setting, HRD status is determined by measuring either evidence for the potential cause or consequence of HRD status. Although patients with germline mutations in *BRCA1/2* are defined as harboring an HRD tumor phenotype, there is a group of approximately 20% of patients that present a positive HRD status based on tumor genomic analyses. For this reason, we selected tumor assays that allowed simultaneous detection of *BRCA* status and genomic instability. It is also worth mentioning that we extensively tested commercial solutions available at the time of our analysis, even before one of them was released to the market, but this is an area of active research and development by vendors, and new kits with improvements are being released from time to time. Nonetheless, to our knowledge, SOPHiA DDM^TM^ HRD solution and AmoyDx^®^ HRD Focus Panel are still the main two commercially available assays deployable in the clinical setting that provide a more comprehensive analysis to determine HRD status. In our analysis, we observed an overall strong correlation across the three assays evaluated, regardless of the different underlying methods employed to assess genomic instability in tumor samples. The highest pairwise correlation observed was between Myriad and SOPHiA, and this assay achieved PPV and NPV of 90.0% and 96.3%, respectively, but we could also observe inter-assay discordances in cases with a PGI score close to the predefined cut-off values. Interestingly, our findings are consistent with previous reports regarding overall agreement scores and success rates among the three commercial assays [36,39].

Despite our results showing that employment of in-house HRD testing is feasible, defining technical procedures and workflows as well as data analysis is not trivial, and many aspects should be considered to achieve successful implementation. The control of the pre-analytical conditions of tumor samples is one of the critical aspects that should be taken into consideration. Inappropriate tissue handling (delayed fixation and over-fixation, inferior paraffin wax quality, and inadequate melting temperature) may modify the quality of the sample, impacting molecular test results. We also anticipate that tumor samples previously submitted to neoadjuvant chemotherapy are more likely to result in an unsatisfactory genomic instability analysis. For NGS-based tumor assays, representative tumor area selection and assessment of the percentage of neoplastic cells, necrosis, and inflammatory components are also important. A minimum of 30% tumor component is recommended to guarantee the detection of a variant through molecular techniques. For some cancers with HRD, this can be difficult to achieve due to abundant inflammatory cell infiltration, and such limitations for the analysis should be clearly stated in the medical report. It is recommended that molecular laboratories and pathology departments maintain quality standards within both pre-analytical and analytical steps. Noteworthy, as shown in our results, when low levels of functional qPCR and insufficient amounts of DNA input are used during library preparation steps, there is an increased chance of low confidence/inconclusive results for the sample being tested. Also, we demonstrated a negative correlation between library yield and residual noise from sequencing experiments. Low-confidence sequencing results because of low DNA library concentration can be related to more fragmented DNA molecules, resulting in potential amplification biases and less uniform coverage in the whole genome.

In our clinical routine procedure, we could significantly reduce the turnaround time with in-house HRD testing. From the test request to the available medical report, it took on average 7 days, whereas the Myriad MyChoice^®^ CDx turnaround time in any diagnostic setting in Brazil is 27 days, being affected by international testing transportation. In addition, the cost of in-house HRD testing corresponds to nearly one-third the price of the Myriad MyChoice^®^ CDx. Considering the clinical need to offer the most effective therapy for the patient in a timely manner, with a widespread testing routine, allowing more patients to benefit from advances in precision medicine, these aspects are crucial to establishing the clinical utility of an assay. However, despite the compelling evidence and advantages of implementing in-house HRD testing, our study presents some limitations. The clinical validity and utility of in-house HRD testing were beyond the scope of this research, and we were unable to contribute with results on this aspect. It is likely that most patients in this study are still undergoing adjuvant treatment and are waiting to start maintenance treatment. Maintenance treatment and follow-up are fundamental parameters, particularly in discordant cases, to assess the clinical outcomes or the benefit of personalized therapy. We were also unable to calculate the failure rate between in-house HRD testing and the reference Myriad MyChoice^®^ CDx assay. This would be valuable to provide a measure of the reliability of the test.

In conclusion, our study shows that the implementation of in-house HRD testing in diagnostic laboratories is technically feasible and can be reliably performed with commercial assays. Also, the turnaround time is compatible with clinical needs, making it an ideal alternative to offer to a broader number of patients while still maintaining high-quality standards at more accessible price tiers. This is the largest HRD testing evaluation using different methodologies and provides a clear picture of the robustness of NGS-based tests currently offered in the market. The present data can be a valuable resource for other clinical laboratories that aim to implement an in-house HRD testing routine.

## Figures and Tables

**Figure 1 diagnostics-13-03293-f001:**
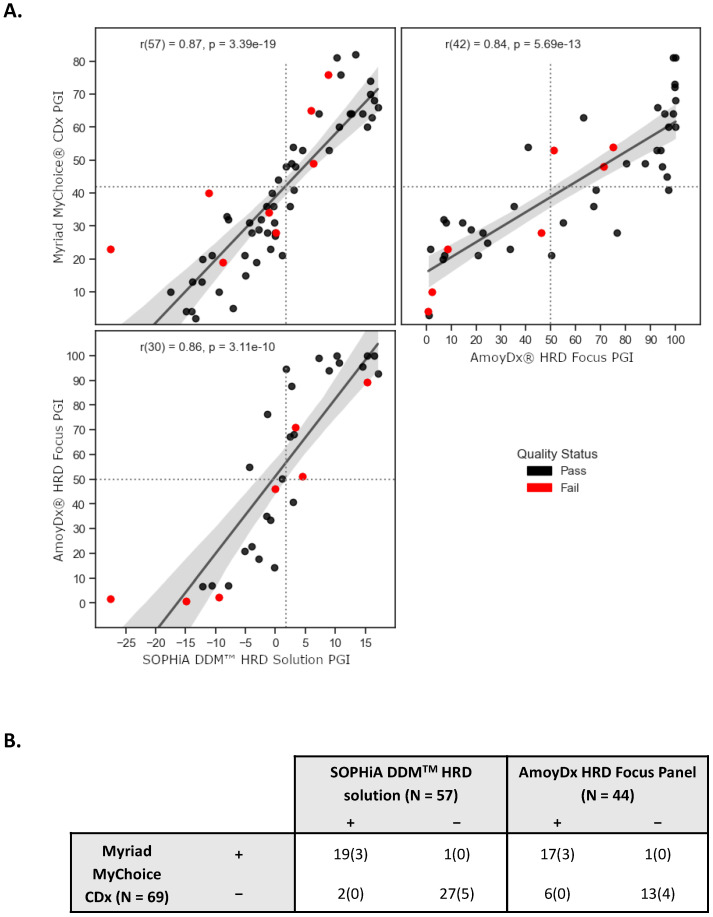
Comparison of predicted genomic instability (PGI) scores across different next-generation sequencing (NGS)-based tumor assays. (**A**) Pairwise correlation of PGI score using three commercially available HRD testing solutions: Myriad MyChoice CDx, SOPHiA DDM^TM^ HRD solution, and AmoyDx HRD Focus Panel. The highest pairwise correlation observed was between Myriad and SOPHiA (R = 0.87, *p*-value = 3.39 × 10^−19^). Bootstrapping-based 95% confidence interval of the regression estimate shown in gray translucent bands around the regression lines. (**B**) Observed discrepancies between the PGI score from Myriad MyChoice CDx and the results obtained from SOPHiA DDM^TM^ HRD solution and AmoyDx HRD Focus Panel. The number of samples that failed quality control (QC) metrics is in parentheses.

**Figure 2 diagnostics-13-03293-f002:**
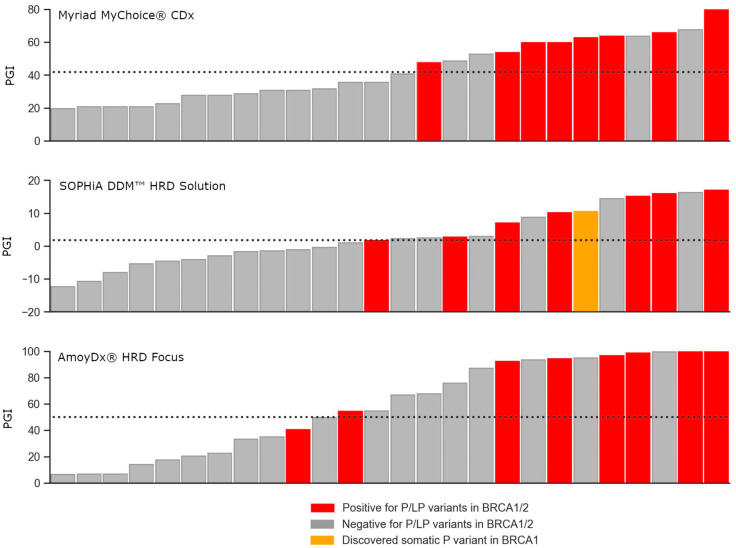
Correlation between *BRCA1/2* status and predicted genomic instability (PGI) score association and consistency between the two biomarkers, *BRCA1/2* status and genomic instability, in a subset of samples that were successfully executed in the three next generation sequencing (NGS)-based tumor assays and passed the quality control (QC) metrics. On top: Myriad MyChoice CDx; in the middle: SOPHiA DDM^TM^ HRD solution; and on the bottom: AmoyDx HRD Focus Panel. The SOPHiA test was able to detect one somatic pathogenic variant in *BRCA1* with a variable allele fraction (VAF) of 9.8%, which was not previously reported by the other two tests. P/LP; pathogenic/likely pathogenic. Bars were ordered by PGI score within each assay, so they do not necessarily follow the same order between assays.

**Figure 3 diagnostics-13-03293-f003:**
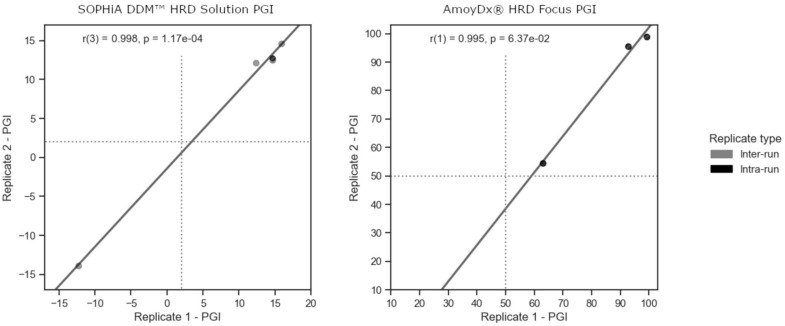
Determination of the precision performance of next-generation sequencing (NGS)-based tumor assays. Reproducibility of intra- and inter-run in-house HRD testing experiments for SOPHiA DDM^TM^ HRD solution and AmoyDx HRD Focus Panel.

**Figure 4 diagnostics-13-03293-f004:**
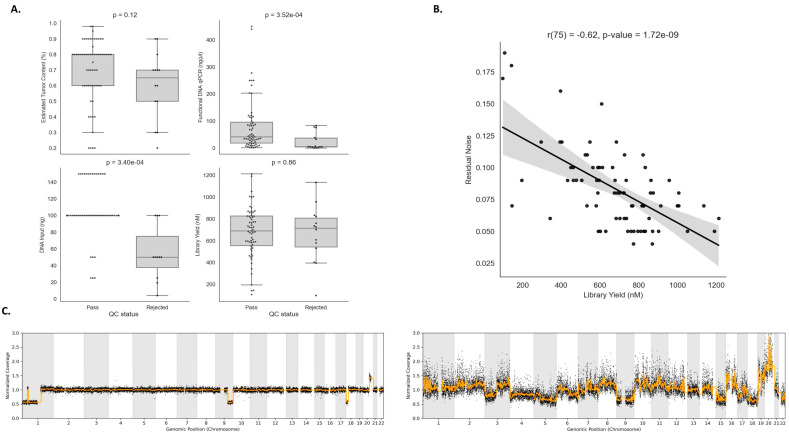
Associations of pre-analytical variables with low-confidence next-generation sequencing (NGS) results. (**A**) Four pre-analytical variables evaluated between pass and low-confidence experiments: (i) estimated tumor content (ETC); (ii) functional DNA qPCR, (iii) DNA input; and (iv) DNA library yield. In our analysis, only functional DNA qPCR and DNA input were statistically significant (*p*-value = 3.52 × 10^−4^; *p*-value = 3.4 × 10^−4^). (**B**) Negative correlation between DNA library yield and residual noise (R = −0.62, *p*-value = 1.72 × 10^−9^). Residual noise is computed by measuring the standard deviation of the normalized genome sequencing coverage profile with respect to the smoothed normalized genome sequencing coverage profile. Bootstrapping-based 95% confidence interval of the regression estimate shown in gray translucent bands around the regression line. (**C**) Two examples of samples that have low (left) and high (right) estimated residual noise profiles.

**Table 1 diagnostics-13-03293-t001:** Clinicopathological features of the tumor samples used in this study.

Characteristics	N
**Number of samples**	85
**Patient age at sample collection, yrs. (Median, IQR)**	60 (19.6)
**Primary tissue**	
Ovary	73
Peritoneum	9
Fallopian tube	3
**BRCA1/2 status**	
Positive for P/LP variants	14
Negative for P/LP variants and VUS	56
Unknown	15
**Estimated tumor content (Median, IQR)**	70% (20%)

P/LP; pathogenic/likely pathogenic; VUS; variants of unknown significance.

**Table 2 diagnostics-13-03293-t002:** In-house HRD testing performance parameters relative to Myriad MyChoice CDx.

	SOPHiA DDM^TM^ HRD Solution	AmoyDx HRD Focus Panel
Positive Predictive Value (PPV)	90.9%	75%
Negative Predictive Value (NPV)	96.3%	100%
Tumors predicted HRD positive	44.9%	64.9%

## Data Availability

All data generated or analyzed during this study are included in this published article.

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
