# Peer review of "A Benchmark of In-House Homologous Recombination Repair Deficiency Testing Solutions for High-Grade Serous Ovarian Cancer Diagnosis"

_diagnostics, 2023, doi:10.3390/diagnostics13213293_

Round 1
Reviewer 1 Report
1. What's the time you conduct this examination in clinical practice?
2. What's the economic benefit of your own examination?
3. What's do you suggest conducting HRD examination in clinical practice in Brazil?
4. Do you think your in-house HRD examination can become global commercially available tool?
Author Response
Please see the file attached below.

Reviewer 2 Report
I read with great interest the manuscript, which falls within the aim of this Journal and offers a high-quality overview of the topic.
Methodology is accurate and conclusions are supported by the data analysis. The discussions sufficiently answer the following questions: main findings of the study, strengths and Limitations of the study, implications and comparison with literature, and future directions.
Although the manuscript can be considered already of high quality, I would suggest taking into account the following minor recommendations:
- I suggest another language revision round to correct a few typos and improve readability.
- I find it interesting to include a reference to the screening programs for early cervical cancer diagnosis (Golia D'Augè T, Giannini A, Bogani G, Di Dio C, Laganà AS, Di Donato V, Salerno MG, Caserta D, Chiantera V, Vizza E, Muzii L, D’Oria O. Prevention, Screening, Treatment and Follow-Up of Gynecological Cancers: State of Art and Future Perspectives. Clin. Exp. Obstet. Gynecol. 2023, 50(8), 160. https://doi.org/10.31083/j.ceog5008160).
- Authors can consider PMID: 37314974 to have some insights.
Considering all these points, I think it could be of interest to the readers and, in my opinion, it deserves the priority to be published pending a few minor revisions.
Minor editing of English language required to make the work clearer and more readable.
Author Response
Please see the file attached below.
